# A Novel ZnO Nanoparticles Enhanced Surfactant Based Viscoelastic Fluid Systems for Fracturing under High Temperature and High Shear Rate Conditions: Synthesis, Rheometric Analysis, and Fluid Model Derivation

**DOI:** 10.3390/polym14194023

**Published:** 2022-09-26

**Authors:** Mahesh Chandra Patel, Mohammed Abdalla Ayoub, Anas Mohammed Hassan, Mazlin Bt Idress

**Affiliations:** 1Department of Petroleum Engineering, Universiti Teknologi Petronas, Perak 32610, Malaysia; 2Petroleum Engineering Department, Khalifa University of Science and Technology, Abu Dhabi 127788, United Arab Emirates

**Keywords:** surfactant-based viscoelastic fluids for fracturing, ZnO nanoparticle assisted viscoelastic fluids, innovative nonpolymeric fracturing fluid compositions, CTAB-based viscoelastic fluids, Herschel–Bulkley fluid models for SBVE or VES, pressure drop estimation during laminar flow of viscoelastic fluids

## Abstract

Surfactant-based viscoelastic (SBVE) fluids are innovative nonpolymeric non-newtonian fluid compositions that have recently gained much attention from the oil industry. SBVE can replace traditional polymeric fracturing fluid composition by mitigating problems arising during and after hydraulic fracturing operations are performed. In this study, SBVE fluid systems which are entangled with worm-like micellar solutions of cationic surfactant: cetrimonium bromide or CTAB and counterion inorganic sodium nitrate salt are synthesized. The salt reagent concentration is optimized by comparing the rheological characteristics of different concentration fluids at 25 °C. The study aims to mitigate the primary issue concerning these SBVE fluids: significant drop in viscosity at high temperature and high shear rate (HTHS) conditions. Hence, the authors synthesized a modified viscoelastic fluid system using ZnO nanoparticle (NPs) additives with a hypothesis of getting fluids with improved rheology. The rheology of optimum fluids of both categories: with (0.6 M NaNO_3_ concentration fluid) and without (0.8 M NaNO_3_ concentration fluid) ZnO NPs additives were compared for a range of shear rates from 1 to 500 Sec^−1^ at different temperatures from 25 °C to 75 °C to visualize modifications in viscosity values after the addition of NPs additives. The rheology in terms of viscosity was higher for the fluid with 1% dispersed ZnO NPs additives at all temperatures for the entire range of shear rate values. Additionally, rheological correlation function models were derived for the synthesized fluids using statistical analysis methods. Subsequently, Herschel–Bulkley models were developed for optimum fluids depending on rheological correlation models. In the last section of the study, the pressure-drop estimation method is described using given group equations for laminar flow in a pipe depending on Herschel–Bulkley-model parameters have been identified for optimum fluids are consistency, flow index and yield stress values.

## 1. Introduction

Hydraulic fracturing technology is a frequently used method for fracturing in low permeability rock formations [1,2]. These fracturing methods have been implemented in the oil industry for more than 40 years [3,4]. So, to induce a fracture and convey the delivered proppant into the fracture, hydraulic fracturing involves injecting a high-pressure fracturing fluid into a reservoir formation [5,6]. This process creates a high formation conductivity in near wellbore zones of fractures [7,8]. Initially in hydraulic fracturing, polymer fluids such as guar gum [9,10] were mainly used as fracturing fluid thickeners [11,12,13,14,15,16,17]. However, traditional polymer-based fluids produce residues, impairing the formation and lower pore conductivity. In addition, only 30 to 45% of the injected guar-based polymer fluids could return from the well during the flow-back period, as shown in a study conducted by Thomas et al. [18,19]. This was caused by the leftover unbroken polymer-based fracturing fluid that obstructed the flow channel [20,21,22]. Moreover, the proppant (sand) can sink to the bottom of the polymer fluids before reaching the fracture tip because of the weak sand suspension capacity. It has been observed that the polymer fluids’ high viscosity can cause fractures to expand in height rather than length [23,24,25].

The fracturing fluid compositions vary based on reservoir rocks and other surrounding factors. The primary function of the fracturing fluid is to fracture the rock and transport proppants in the fracture. So, the fluids should be able to carry and transport sand proppants from surface facilities to the newly created fractures in the subsurface and then break them down so that the proppants can be settled in the fracture gaps. At the same time, the remaining fluid should flow back to the surface. The conventional polymeric fracturing fluids have many issues, such as polymeric and crosslinker residue in the formation, which leads to damage [15], substantial amount of trapped water, etc. The surfactant-based viscoelastic (SBVE) fluids are deemed capable of eliminating these issues and emerging as an environmentally friendly green technique for fracturing. Since Schlumberger presented the concept of viscoelastic surfactants (VES) or SBVE fluid as a thickening agent for fracturing fluid in 1997 [26,27,28], the viscoelastic behavior of SBVE fluids and the no makeup of the residue after the gel breaks have made them an appealing approach in the oil and gas industry [29,30,31].

Not only good viscosity, but also having a good elasticity enable SBVE fluids to be a perfect alternative candidate to transport proppants [26]. The worm-like micelles (WLMs) that are responsible for this viscoelastic behavior [27]. These viscoelastic WLMs are smart self-organized structures that can be applied in a wide range of oil and gas industry operations, such as hydraulic fracturing, emulsions, polymer, surfactant, and foam flooding [9,31,32].

However, these cylindrical micelles are highly susceptible to hydrocarbons. During the completion stage of the hydraulic fracturing operation, the carrier liquid will be destroyed by the influence of the formation hydrocarbon and can be easily removed from the fractures. Consequently, the high permeable path of the fracture will be achieved for the formation fluids to flow. Nonetheless, application of SBVE fluids at high temperature conditions in deep wells is a huge challenge [33]. The viscous stability of fracturing fluids with respect to the temperature and shear rate changes are key parameters to consider, which determine the proppants’ carrying potential of the fluid [30,34,35].

Therefore, developing improved viscoelastic systems of SBVE fluids using other additives is necessary [11,12,13]. These can provide a high elasticity modulus and viscosity stability at elevated temperatures and moderate filtrate recovery [14,15,16,36]. In addition, WLMs that react to external stimuli are being researched to control viscoelastic behavior better and understand its applicability under different environments [37,38,39].

Over the last few years, researchers have realized that nanoparticles can improve surfactant-based viscoelastic fluids’ performance. The nanoparticles establish electrostatic bridges to surfactant micelles which modify the microstructural behaviours and rheology of the viscoelastic fluid system. The nanoparticles strengthen the entanglements of worm-like micelles providing increased micellar length and consequently improving rheological characteristics such as viscosity [30,40,41,42,43]. The viscoelastic fluid systems consistently showed improved properties and stability under adverse conditions when metal oxide nanoparticles were added [40,41,42,43].

Cetyltrimethylammonium bromide (CTAB) are quaternary ammonium halides that make spherical micelles after a critical micellar concentration. The micelles of these surfactants grow from spherical to rod-shaped by adding of different counter-ions [8,37]. Generally, Halide anions associate with surfactant headgroups moderately with gradual micellar growth. However, with specific anions that associate strongly, such as inorganic and aromatic salt reagent anions (e.g., NO_3_^−^ of Sodium Nitrate), the surfactant solutions give a remarkable viscosity increase due to rapid growth in rod-shaped micelles even at low surfactant and salt concentrations [30,38,43].

For instance, Chieng, Z. H., et al. [44] reported mixing organic acids, citric acid (CA) and maleic acid (MA) at respective molar ratios of (3:1) and (2:1), with long chain cationic surfactant cetyltrimethylammonium bromide (CTAB). This was a novel way to create a CTAB-based VES-fluid solution with the optimum fracture capabilities. Experimental confirmation of the CTAB-based VES-thickening fluid’s viscoelastic behavior at a temperature of 90 °C demonstrated CTAB-CA VES-fluid as desirable thickening fracturing fluid [44].

In this experimental study, SBVE fluids were synthesized using cationic surfactant CTAB and sodium nitrate (NaNO_3_) as counter ion salt reagents. Different SBVE fluids are synthesized at a fixed surfactant concentration (0.1 M) and different salt reagent concentrations. This category of fluid is termed type1 fluids. No author has studied this composition of SBVE fluids previously to implement them for hydraulic fracturing purposes. The rheological characteristics: viscosity and shear stress have been analysed using a rotational rheometer by varying shear rates from 1 to 500 sec ^−1^ at 5 sec ^−1^ intervals and different temperatures which is a novel approach to understand the ability of SBVE fluids under fiend-like conditions during fracturing. The authors found a massive drop in viscosity at high temperatures and at high shear rates (HTHS) conditions. Therefore, they investigated other possible ways to improve the SBVE fluid system.

Recently, some studies have been conducted on the application of zinc oxide nanoparticles (ZnO NPs) for drilling fluid compositions under different conditions especially under high temperature conditions. The studies reported that ZnO NPs enhance the fluid properties by providing stable viscosity, less fluid loss, inhibitive nature, and ability to remove H_2_S [45,46,47].

Therefore, the authors chose to investigate ZnO NPs with SBVE considering them to be a potential candidate for improvements in rheological characteristics of the synthesized SBVE fluid system. The authors hypothesized that ZnO nanoparticles could improve this fluid’s rheological characteristics under HTHS conditions. Therefore, the next version of fluids is synthesized by implementing nanofluids of zinc oxide (ZnO NPs dispersion in water) were termed type2 fluids. The viscosity plots for all fluids of type1 and type2 categories were analysed at a fixed temperature of 25 °C by varying shear rate values to identify optimum fluids with the highest viscosity values for the entire shear rate range.

The rheology of optimum fluids compared for all shear rate values (1 to 500 Sec^−1^) at different temperatures of 25 °C, 35 °C, 45 °C, 55 °C, 65 °C and 75 °C respectively to visualize the effect of increasing shear rate conditions at different temperatures. The descriptive plots depict viscosity at each temperature for the range of shear rates which helps to check change in rheological characteristics due to ZnO NPs additives at each case of HTHS.

Further, the authors have developed Herschel–Bulkley fluid models for synthesized viscoelastic fluids systems depending on statistical analysis and correlation parameters identified as consistency, flow index and yield stress on plotted rheometric parameters: viscosity and shear stress values with varying shear rate values at 25 °C temperature conditions. In the last section of this study, the pressure-drop estimation method described using given group equations for laminar flow in a pipe depending on Herschel–Bulkley-model parameters will be identified for optimum fluids in both categories.

## 2. Materials and Methods

Cetyltrimethylammonium bromide, cetrimonium bromide, hexadecyltrimethylammonium bromide or CTAB is a quaternary ammonium surfactant. It is one of the components of topical antiseptic cetrimide, and its molecular structure is illustrated in Figure 1. The chemical formula for CTAB is ([(C_16_H_33_)N(CH_3_)_3_] Br) with a molecular weight of 364.447 gm/mol. The cationic surfactant CTAB (98% pure) of Loba Chemie Pvt. Ltd. was obtained from Sigma Aldrich. AR grade Sodium Nitrate salt (with a minimum assay of 99%) for anionic nitrate counter ion was obtained from ACS chemicals (Molecular structure in Figure 1). The nanofluid of ZnO dispersion nanoparticles (<100 nm particle size TEM), 20 wt% in H_2_O was obtained from sigma Aldrich.

The aqueous solutions of cationic surfactants such as hexadecyltrimethylammonium bromide (CTAB) form long worm-like micelles (WLMs) upon adding specific salts, strongly binding counter-ions or cosurfactants. The enthalpy of micellization and Gibs free energy for micellization seems to be the lowest for NO_3_^−^ [17,48] compared with other inorganic anions, as reported by Jiang et al. (2005) [48], this indirectly indicates the entropy of micellization in CTAB solution. Earlier, K. Kuperkar et al. (2008) [38] investigated viscoelastic solutions of (WLMs) formed in aqueous solutions of the cationic surfactant CTAB in the presence of the salt reagent NaNO_3._ They reported that the addition of NaNO_3_ to CTAB micelles leads to a decrease in the surface charge of the ellipsoidal micelles and, thus, an increase in their length occurs. Researchers have also reported that NaNO_3_ is a highly effective inorganic electrolyte to induce worm-like micelle (WLMs) formation and branching in the micellar solution of Cetyltrimethylammonium bromide (CTAB) [38].

### 2.1. Preparation of Type1 Fluids without Nanoparticle Additives

The surfactant solution was prepared with a fixed CTAB concentration of 0.1 M in demineralized water, and different viscoelastic fluids were prepared by varying salt concentrations from 0.2 M to 2.0 M.

The transparent surfactant solution (d) was prepared by ultrasonication bathing (c) of white solution of demineralized solvent water (a) and solute cetrimonium bromide (b), as shown in Figure 2. Then the inorganic sodium nitrate salt reagent was added, and the prepared solution (e) was mixed by heating and stirring (f) using a magnetic stirrer. The prepared fluid goes through an ultrasonic bath (g) which removes air bubbles in the fluid, and a homogeneous viscoelastic surfactant fluid (h) fluid is prepared.

### 2.2. Preparation of Type2 Fluid with Nanoparticle Additives

Initially, the 1 wt% ZnO NP dispersion nanofluid solution was prepared using demineralized water and adding 20 wt% ZnO nanoparticle dispersion fluid (in water). The surfactant solution was prepared with a fixed 0.1 M CTAB concentration and nanofluid solution. Then different viscoelastic fluids were prepared by varying salt concentrations from 0.2 M to 2 M.

Initially, a homogenous 1% weight of ZnO nanoparticle dispersion in water (1 wt% ZnO nanofluid) (a) was prepared using demineralized water and 20% weight ZnO nanoparticle dispersion in water obtained from Sigma Aldrich(Gujarat, India). The prepared fluid goes through an ultrasonic bath (h) which removes air bubbles in the fluid, and a homogeneous viscoelastic surfactant fluid (i) is prepared. The slightly white surfactant solution (e) was prepared by ultrasonication bathing (d) of white solution of solvent nanofluid (b) and solute Cetrimonium bromide (c). Then the inorganic sodium nitrate salt reagent was added, and the prepared solution (f) was mixed by heating and stirring (g) using a magnetic stirrer, as shown in Figure 3.

## 3. Rheological Characterization and Observations

The focus of this research is to investigate the ability of nanoparticles to improve the viscoelastic fluid by improving its rheological characteristics. The rheometric characterization and analysis of the synthesized viscoelastic fluids were performed using the Anton-Par rotational rheometer (MCR2Model). Then, the viscosity was observed by varying shear rates from 1 to 500 Sec^−1^ at a difference of 5 Sec^−1^ at different temperature conditions from 25 °C to 75 °C at 10 °C intervals.

### 3.1. Analysis of Type1 Fluids and Optimization of Salt Concentration

The viscosity data with increasing NaNO_3_ salt reagent concentration at constant temperature and increasing shear rate conditions (1 to 500 Sec^−1^) were plotted at 25 °C. The graphical plots depict that with increasing salt concentration, the viscosities of the fluids increases until 0.8 M NaNO_3_ salt concentration and give similar viscosities for 1.0 M salt concentration, see Figure 4. Further, as the salt concentration increased to 1.5 M and 2.0 M NaNO_3_ concentration, the plot showed a decrease in viscosity values. So, 1.0 M and 0.8 M concentration are candidates for optimum concentration. We can decide the optimum concentration depending on maintained better rheology or viscosity of the fluid at elevated temperature and shear rate conditions. (See Figure 5)

The statistical analysis of the plotted data gives different correlations between viscosity and shear rate with a considerable coefficient of determination values (see Table 1).

Figure 5 depicts that the viscosity values of the fluids at 0.8 M and 1.0 M seem to be similar at 25 °C temperature for all shear rate ranges. However, as the temperature increases, the viscosity values are higher for 0.8 M concentration fluid for the entire range of shear rates at all temperatures. Thus, 0.8 M concentration fluid can be considered the optimum for the type1 fluids category (See in Appendix A).

To understand the effect of temperature, the rheology data of optimum fluid at 0.8 M salt concentration were plotted at different temperatures at 25 °C, 35 °C, 45 °C, 55 °C, 65 °C and 75 °C for the range of shear rates from 1 to 500 sec^−1^ (See Figure 6). A decrease in viscosity values were found as the temperature increased for the entire shear rate range.

### 3.2. Analysis of Type2 Fluids and Optimization of Salt Concentration

Here the viscosity increases initially with increasing salt concentration from 0.2 M to 0.4 M and 0.6 M, as seen in Figure 7. Then it starts to decrease with increasing salt concentration. Then, as the salt concentration increases, the viscosity plots decrease. There is not much difference between values at 0.4 and 0.6 M concentration, but at 0.6 M, it shows a better viscosity at an even higher shear rate. Again, similar to type1 fluids, 0.8 M and 0.6 M were compared for all temperature conditions (see Figure 8) to identify the optimum concentration. The fluid3 of 0.6 M salt concentration showing higher viscosity values on the entire range of shear rates and temperatures, indicating that fluid3 of 0.6 M is an optimum fluid for type2 categories.

Similar to type1 fluids, the statistical analysis of the plotted data of type2 fluids also give different correlations between viscosity and shear rate with a considerable coefficient of determination values. (See Table 2)

The temperature effect on rheology of optimum fluid 0.6 M salt concentration in type2 category fluids has been illustrated in Figure 9.

Here the plot depicts that with increasing temperature, the rheology plots remain almost similar up to 55 °C. At temperatures of 65 °C and 75 °C, the viscosity decreases, as seen in Figure 9 which displays a different trend than type1 fluids where viscosity decreases continuously as the temperature condition changes (see Figure 6).

### 3.3. Comparison of the Rheology of Type1 and Type2 Fluid Categories

The rheology in terms of viscosity of optimum viscoelastic fluids of type1, which are without NPs additives and type2 which are synthesized using nanofluid of ZnO NPs dispersion has been compared in Figure 10. The figure depicts that type2 optimum fluid (with ZnO NP additives) shows better rheology with increasing shear rate and temperatures conditions than type1 optimum fluid (seen Figure 10).

Similarly, the viscosity values of the same salt concentration fluids of 0.6 M for both type1 and type2 categories were compared (see Figure 11). It is observed that the difference is much higher than in optimum fluids (see Figure 10). As it is the same concentration of salt, we can conclude that ZnO nanoparticles assist maintaining the entangled structure of WLM at HTHS conditions.

### 3.4. Shear Stress Plots and Yield Stress Analysis for Type1 and Type2 Fluids

Figure 12 depicts shear stress plots for the type1 fluids. The shear stress vs shear rate curves was almost similar for values of 0.6 M to 1.5 M NaNO_3_ concentration. The type1 fluid6 of 1.5 M NaNO_3_ concentration and type1 fluid4 of 0.8 M NaNO_3_ concentration has the highest yield stress of 28, as shown in Table 3.

Figure 13 depicts shear rate versus shear stress plots based on rheometric analysis data of type2 SBVE fluids with varying NaNO_3_ salt concentration at 25 °C. The maximum yield stress value was 40 Pascal, represented by fluid3 of 0.6 M NaNO_3_ concentration, as demonstrated in Table 4.

The yield stress characteristic of complex fluids or non-Newtonian fluids is a property associated with the material not flowing unless the applied stress exceeds a specific value. The yield stress is the stress value that must be applied to the sample before it starts to flow. Similarly, like stretching a spring, the sample deforms elastically below the yield stress; above the yield stress, the sample flows like a liquid [38,49].

The Figure 14 shows the shear stress of a complex fluid which appears to have yield stress but shows viscous behavior at much lower shear rates. This is similar case for SBVE fluids showing rubber-like elastic behavior below the yield stress.

Table 3 and Table 4 below enlist the yield stress values for type1 and type2 fluids as identified in Figure 12 and Figure 13.

## 4. Fluid Modelling and Pressure Drop during Laminar Flow in Pipeline

Winslow Herschel and Ronald Bulkley’s model introduced the model of non-Newtonian fluids in 1926, in which the strain experienced by the fluid is related to the stress in a complicated and non-linear way. The relationship is characterized by three parameters which are the consistency *k*, the flow index *n*, and the yield shear stress *τ*_0_. The flow index measures the degree to which the fluid is shear-thinning or shear-thickening, and the consistency is a simple constant of proportionality [50,51]. The yield stress quantifies the amount of stress the fluid may experience before it yields or deforms and begins to flow.

We can estimate consistency value k and flow index value n by analyzing statistical regression functions of all viscoelastic fluids of type1 and type2.

The constitutive equations of the Herschel–Bulkley model after the yield stress have been reached and can be written as follows (Equations (1) and (2)) [52,53,54]
(1)$$\tau =\tau_{0}+k{ϓ}^{n}\;\mathrm{for}\;\tau \;\geq \;\tau_{0}$$

And
(2)$$\eta =\tau_{0}{\left|ϓ\right|}^{-1}+k{\left|ϓ\right|}^{n-1}\;\mathrm{for}\;\tau \;\geq \;\tau_{0}$$

Here *𝜏* is shear stress values in Pa, *𝜏*_0_ is the yield stress value in Pa, ϓ is shear rate values in Sec^−1^, *k* is fluid consistency, *n* is flow index and *η* is viscosity in Pa-Second or Centipoise.

Therefore, the constitutive equation of the Herschel–Bulkley model after the yield stress has been reached for optimum fluids can also be estimated.

The Herschel–Bulkley model equation (after yield stress has been reached) for the optimum fluid of type1 category fluids without nanoparticle additives, fluid with 0.8 M NaNO_3_ concentration can be expressed as below, where the values have been taken from Table 3 and Table 5.
(3)$$\tau =28+9.714.3{ϓ}^{0.36}$$
(4)$$\eta =28\times {\left|ϓ\right|}^{-1}+9.714.3\times {\left|ϓ\right|}^{-0.64}$$

Similarly, the Herschel–Bulkley model equation (after yield stress has been reached) for the optimum fluid of type2 category fluids with nanoparticle additives, fluid with 0.6 M NaNO_3_ concentration can be expressed as below. The values have been taken from Table 4 and Table 6.
(5)$$\tau =40+11.351{ϓ}^{0.394}$$
(6)$$\eta =40\times {\left|ϓ\right|}^{-1}+11.351\times {\left|ϓ\right|}^{-0.606}$$

Chilton and Stains represented a set of equations (Equations (7)–(11)) to calculate the pressure drop for laminar flow for such fluids [55]. The equations require an iterative method to extract the pressure drop, as it is present on both sides of the equation [52,55].
(7)$$\frac{∆P}{L}=\frac{4k}{D}{\left(\frac{8V}{D}\right)}^{n}{\left(\frac{3n+1}{4n}\right)}^{n}\frac{1}{1-X}\left(\frac{1}{1-aX-bX^{2}-cX^{3}}\right)$$
(8)$$X=\frac{4L\;\tau_{0}}{D∆P}$$
(9)$$a=\frac{1}{2n+1}$$
(10)$$b=\frac{2n}{\left(n+1\right)\left(2n+1\right)}$$
(11)$$c=\frac{2n^{2}}{\left(n+1\right)\left(2n+1\right)}$$

Here *P* is the Pressure Drop in Pa, *L* is the pipe length in meters, and *D* is the diameter of the pipe in meters.

Therefore, as suggested by Chilton and Stains, the pressure drop during laminar flow in a pipe can be estimated for both optimum fluids of type1 and type2 groups using factors calculated *a*, *b*, and *c* in Table 7 and implementing an iterative method as suggested by the authors [55].

## 5. Discussion

Micellar solution of surfactants having wormlike micelles or cylindrical micelles changes to viscoelastic fluids with good rheological characteristics under certain conditions due to entanglements in micelles. These viscoelastic systems are sensitive to changes in conditions. So, these viscoelastic systems are not able to maintain rheology under high temperature and high shear rate (HTHS) conditions. The nanoparticle additives can assist to maintaining or improving their rheology even at HTHS conditions.

In this study, initially, a micellar solution of cationic surfactant of cetrimonium bromide or CTAB was prepared which had long cylindrical or worm-like micelles. The micellar solution formed a highly viscous, viscoelastic fluid system in the presence of counterion sodium nitrate salt reagents due to entanglements in long WLMs of CTAB. The rheology was analyzed using a rotational rheometer with varying temperature and shear rates. The represented rheology of the fluids was found to be high enough that the fluids can be implemented successfully on the field for hydraulic fracturing operations (see in the Appendix A). These SBVE fluids leave no residual of polymers or crosslinkers in the formations near the fractured area. Therefore, these SBVE fluids have the ability to avoid formation damage near fractured area.

The rheology of the fluids changes with varying concentrations of the salt reagent. The viscosities show high values up to certain concentrations of the counter ion salt reagent for both fluid categories with and without ZnO nanoparticles additives: type1 and type2 fluids. Beyond which the fluids represent lesser viscosity values when increasing the salt concentration, which is same for both the fluid categories. Hence, the optimum fluid concentrations have been identified as 0.8 M NaNO_3_ salt concentration for type1 fluids and 0.6 M NaNO_3_ for type2 fluids which show the highest viscosity for all shear rate and temperature conditions. The rheological characteristics have been analyzed using a rotation rheometer. The viscosity values for both types of fluids decrease with increasing the shear rates and temperature conditions. However, the authors hypothesized that ZnO NPs would improve the rheology of type1 fluids by supporting entangled WLMs structures, which is proven true. The average viscosity comparison of optimum fluids at 25 °C and different shear rate ranges has been illustrated in Figure 15.

The viscosity values of the type2 optimum fluid (0.6 M salt reagent concentration) with ZnO nanoparticles additives are higher when compared to type1 (without nano additives) fluids of 0.6 and 0.8 (optimum) salt concentration for entire range of shear rates (1 to 500 Sec^−1^) at all temperature conditions: 25 °C, 35 °C, 45 °C, 55 °C, 65 °C and 75 °C respectively as illustrated in Figure 10 and Figure 11. The authors analyzed the rheometric data to get correlations and fluid and pressure drop equation models.

The effect of temperature on the rheology of synthesized optimum SBVE fluids is illustrated in Figure 6 and Figure 9, which show that viscosity values at any constant shear rate decreased gradually with an increase in temperature of type1 (without nano additives) optimum fluid system while it remains similar up to 55 °C in the case of type2 (with ZnO nano additives) optimum fluid and shows lesser viscosities at 65 °C and 75 °C. Therefore, the ZnO nanoparticle additives have the ability to maintain the rheology of the SBVE system with increasing temperature conditions up to 55 °C.

The statistical analysis and correlations of rheological characteristics with varying shear rates at 25 °C proves that the fluid follows Herschel–Bulkley fluid models. Flow index, consistency and yield stress were identified to characterize the fluids. Subsequently, method and equation models are suggested for the estimation of pressure drop during laminar flow in a pipe depending on the identified characteristic parameters of Herschel–Bulkley models. These models and methods will help to understand the behaviour of SBVE fluids during their on-field implementations for hydraulic fracturing purposes. However, these SBVE fluid systems should be investigated more profoundly before their on-field implementations considering other aspects of hydraulic fracturing operations, such as how the fluids behave with different rock mineralogy, what type of oil and gas formations are best suited for these fluids and what other compositions (such as breakers, friction reducers etc.) can be added to them to cover remaining important technical aspects for a successful on-field applications during hydraulic fracturing operations.

## Figures and Tables

**Figure 1 polymers-14-04023-f001:**
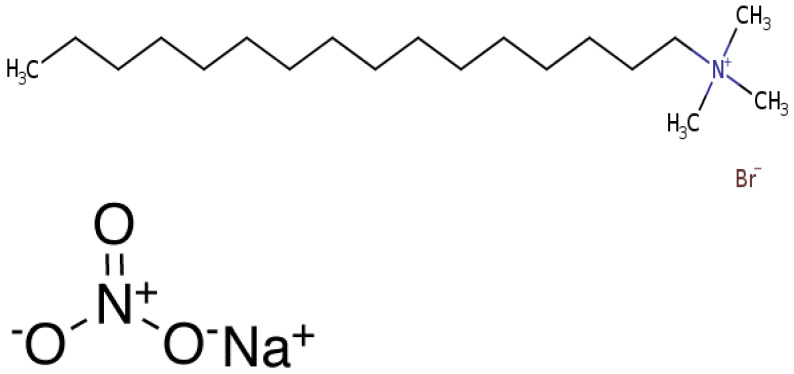
Molecular structure of CTAB and NaNO_3_.

**Figure 2 polymers-14-04023-f002:**
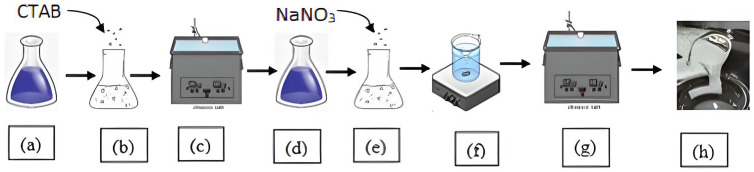
Process of preparation of type1 viscoelastic fluid without nanoparticle additives; Steps (**a**–**c**) show the process for preparation of surfactant solution and steps (**d**–**h**) depict the process of formation of SBVEF; steps (**c**,**g**) show ultrasonication process and step (**f**) shows magnetic stirring.

**Figure 3 polymers-14-04023-f003:**
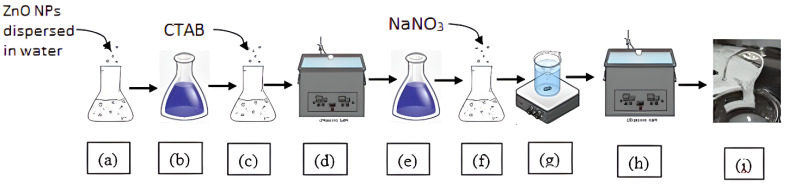
Process of preparation of type2 viscoelastic fluid with nanoparticle additives; Steps (**a**–**d**) show the process for preparation of surfactant nanofluid solution and steps (**e**–**i**) depict the process of formation of SBVEF; steps (**d**,**h**) show ultrasonication process and step (**g**) shows magnetic stirring.

**Figure 4 polymers-14-04023-f004:**
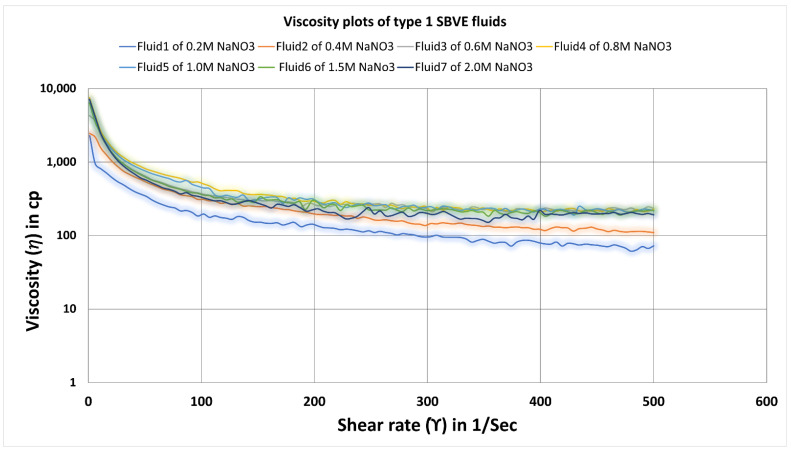
Viscosity curves for SBVE fluids of type1 category with varying NaNO_3_ concentration at 25 °C.

**Figure 5 polymers-14-04023-f005:**
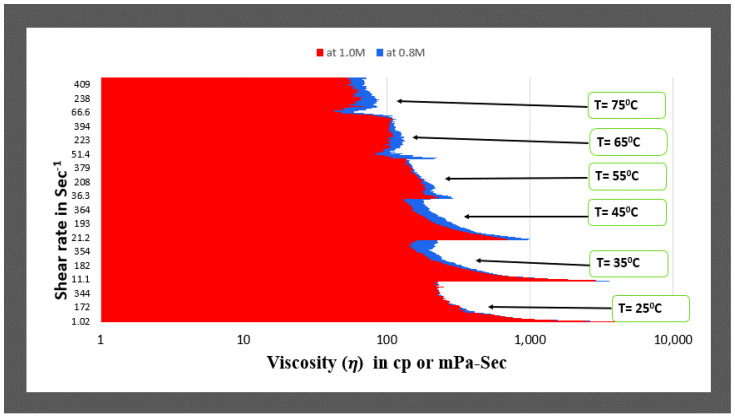
Comparison of viscosity plot of type1 viscoelastic fluids containing 0.8 M and 1.0 M NaNO_3_ concentration at different temperatures.

**Figure 6 polymers-14-04023-f006:**
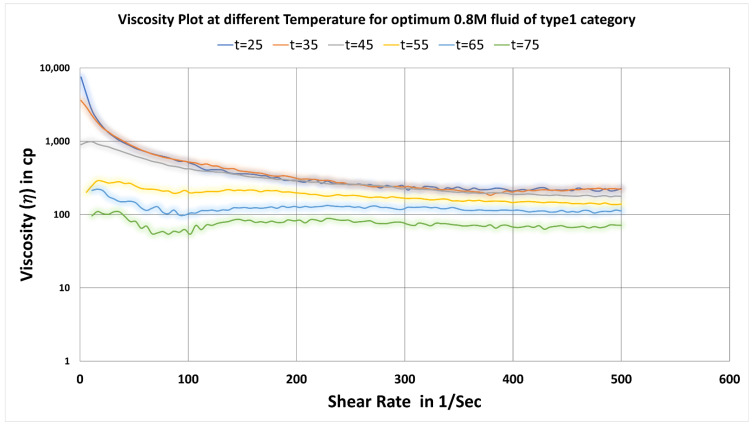
Change in viscosity plot of optimum concentration (0.8 M) of type1 fluids at varying temperatures.

**Figure 7 polymers-14-04023-f007:**
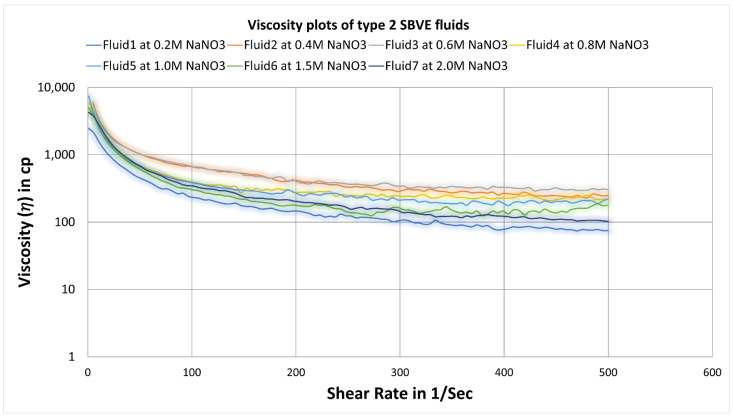
Viscosity curves for surfactant-based viscoelastic fluids of type2 with varying NaNO_3_ concentrations at 25 °C. Please change the same as below.

**Figure 8 polymers-14-04023-f008:**
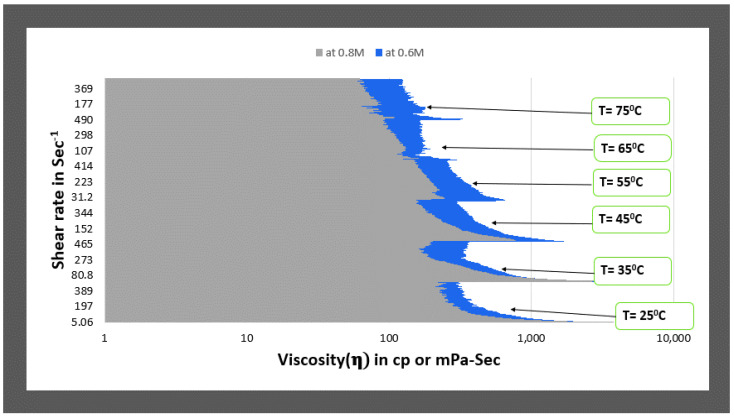
Comparison of viscosity plot of type2 viscoelastic fluids containing 0.6 M and 0.8 M NaNO_3_ concentration at different temperatures.

**Figure 9 polymers-14-04023-f009:**
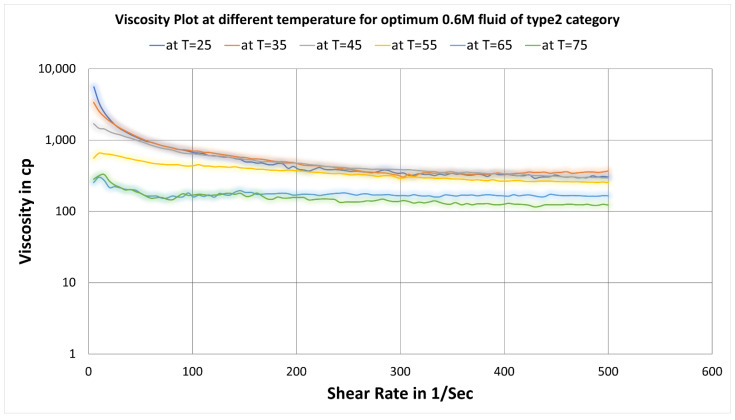
Change in viscosity plot of optimum concentration (0.6 M) fluid of type1 fluids at varying temperatures.

**Figure 10 polymers-14-04023-f010:**
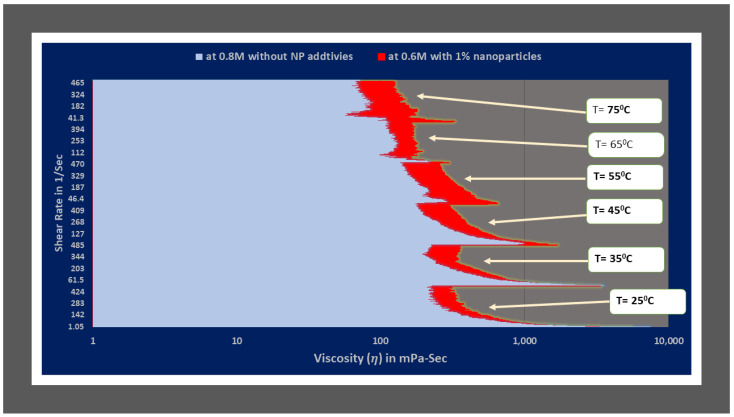
Rheology comparison in terms of viscosity of optimum viscoelastic fluids of type1 (without NP additives) and type2 (with NP additives) categories.

**Figure 11 polymers-14-04023-f011:**
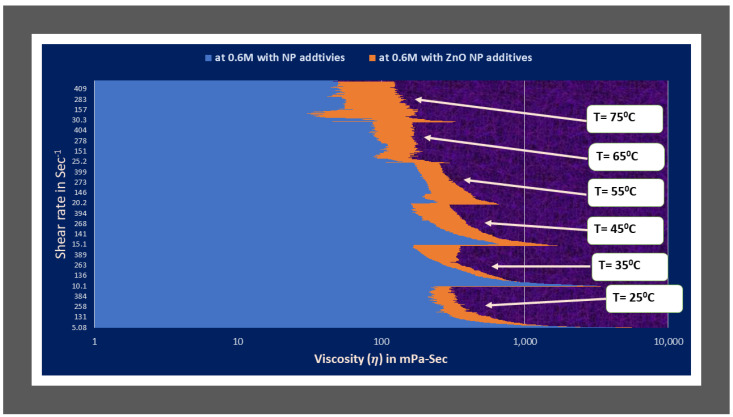
Rheology comparison in terms of viscosity of optimum viscoelastic fluids of type1 (without NP additives) and type2 (with NP additives) fluids at 0.6 M NaNO_3_.

**Figure 12 polymers-14-04023-f012:**
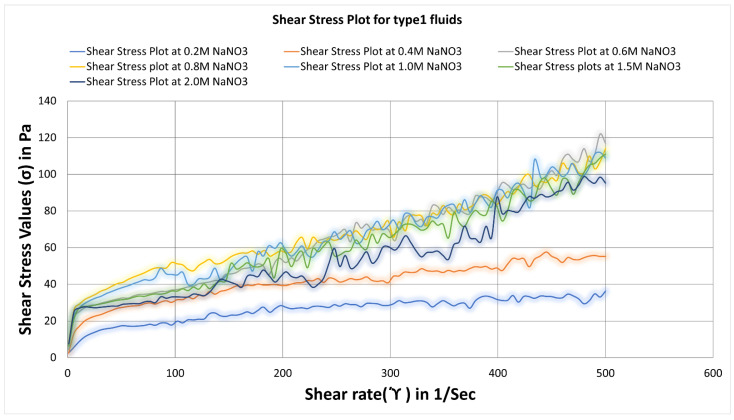
Shear stress vs shear rate curves for surfactant-based viscoelastic fluids of type1 with varying NaNO3 concentrations at 25 °C.

**Figure 13 polymers-14-04023-f013:**
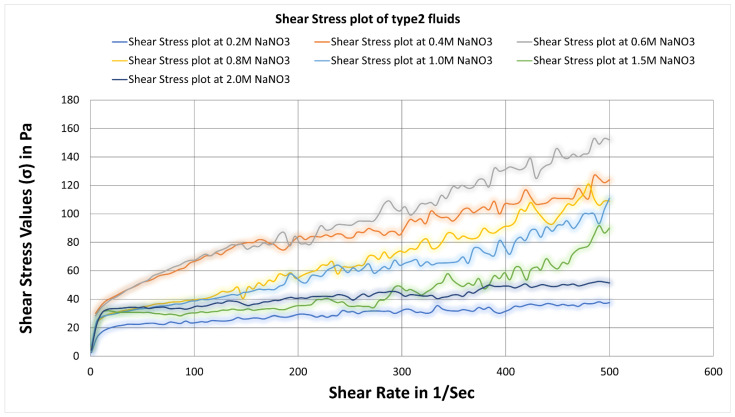
Shear stress vs shear rate curves for surfactant-based viscoelastic fluids of type2 with varying NaNO_3_ concentrations at 25 °C.

**Figure 14 polymers-14-04023-f014:**
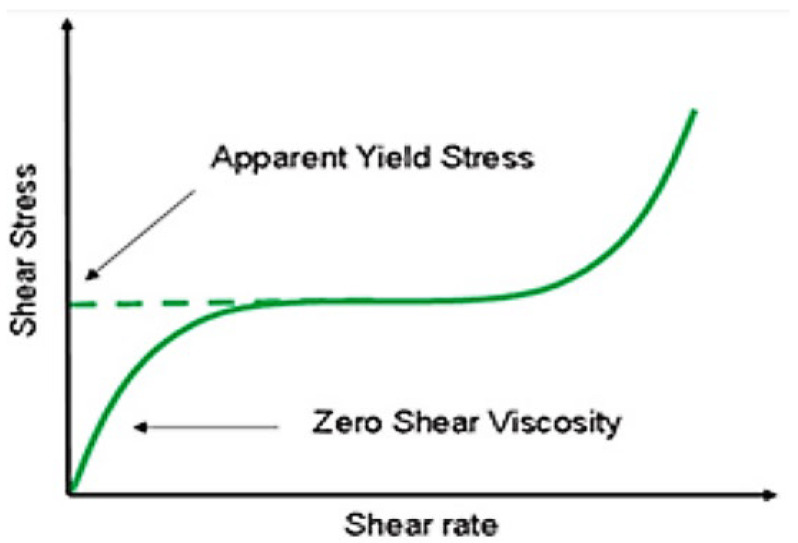
A Plot of shear stress of material that appears to have yield stress but shows viscous behavior at much lower shear rates [50].

**Figure 15 polymers-14-04023-f015:**
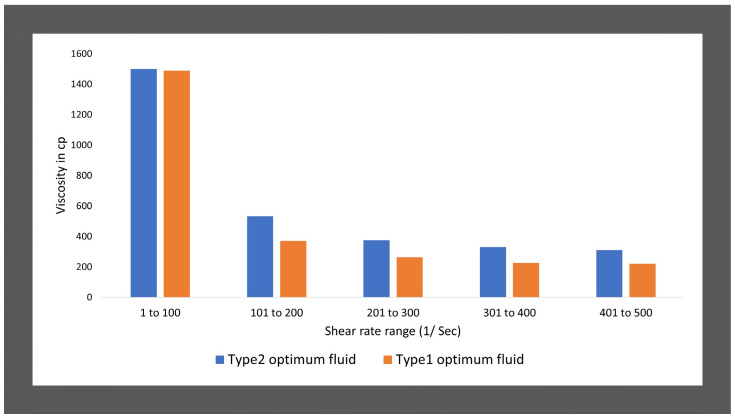
Average viscosity comparison of optimum fluids of type1 and type2 categories at 25 °C for different shear rate ranges.

**Table 1 polymers-14-04023-t001:** Regression functions for viscosity change with a varying shear rate of different viscoelastic type1 fluids at Temperature 25 °C.

Sr. No.	Fluid	Regression Function	Statistical Coefficient of Determination (R^2^ Value)
1	Fluid 1 of 0.2 M NaNO_3_ Concentration	*η* = 3.7839ϓ^−0.641^	R^2^ = 0.9611
2	Fluid 2 of 0.4 M NaNO_3_ Concentration	*η* = 9.1389ϓ^−0.703^	R^2^ = 0.8954
3	Fluid 3 of 0.6 M NaNO_3_ Concentration	*η* = 8.9884ϓ^−0.637^	R^2^ = 0.961
4	Fluid 4 of 0.8 M NaNO_3_ Concentration	*η* = 9.7143ϓ^−0.64^	R^2^ = 0.9418
5	Fluid 5 of 1 M NaNO_3_ Concentration	*η* = 7.7744ϓ^−0.604^	R^2^ = 0.9465
6	Fluid 6 of 1.5 M NaNO_3_ Concentration	*η* = 6.4291ϓ^−0.586^	R^2^ = 0.9559
7	Fluid 7 of 2 M NaNO_3_ Concentration	*η* = 6.8167ϓ^−0.619^	R^2^ = 0.9492

**Table 2 polymers-14-04023-t002:** Regression functions for viscosity change with a varying shear rate of different viscoelastic type2 fluids at a temperature of 25 °C.

Sr. No.	Fluid	Regression Function	Statistical Coefficient of Determination (R^2^ Value)
1	Fluid 1 of 0.2 M NaNO_3_ Concentration	*η* = −0.324 ln(ϓ) + 1.9278	R^2^ = 0.8217
2	Fluid 2 of 0.4 M NaNO_3_ Concentration	*η* = 15.340ϓ^−0.68^	R^2^ = 0.9933
3	Fluid 3 of 0.6 M NaNO_3_ Concentration	*η* = 11.351ϓ^−0.606^	R^2^ = 0.9824
4	Fluid 4 of 0.8 M NaNO_3_ Concentration	*η* = 6.5698ϓ^−0.573^	R^2^ = 0.9421
5	Fluid 5 of 1 M NaNO_3_ Concentration	*η* = 7.8174ϓ^−0.624^	R^2^ = 0.9559
6	Fluid 6 of 1.5 M NaNO_3_ Concentration	*η* = 8.2571ϓ^−0.691^	R^2^ = 0.8039
7	Fluid 7 of 2 M NaNO_3_ Concentration	*η* = −1.085 ln(ϓ) + 5.0358	R^2^ = 0.9546

**Table 3 polymers-14-04023-t003:** Yield stress values for type1 fluids at 25 °C.

Sr. No.	Fluid	Yield Point in Pa
1	Fluid 1 of 0.2 M NaNO_3_ Concentration	14
2	Fluid 2 of 0.4 M NaNO_3_ Concentration	22
3	Fluid 3 of 0.6 M NaNO_3_ Concentration	24
4	Fluid 4 of 0.8 M NaNO_3_ Concentration	28
5	Fluid 5 of 1 M NaNO_3_ Concentration	27
6	Fluid 6 of 1.5 M NaNO_3_ Concentration	28
7	Fluid 7 of 2 M NaNO_3_ Concentration	27

**Table 4 polymers-14-04023-t004:** Yield stress values for type2 fluids at 25 °C.

Sr. No.	Fluid	Yield Point in Pa
1	Fluid 1 of 0.2 M NaNO_3_ Concentration	18
2	Fluid 2 of 0.4 M NaNO_3_ Concentration	39
3	Fluid 3 of 0.6 M NaNO_3_ Concentration	40
4	Fluid 4 of 0.8 M NaNO_3_ Concentration	25
5	Fluid 5 of 1 M NaNO_3_ Concentration	27
6	Fluid 6 of 1.5 M NaNO_3_ Concentration	32
7	Fluid 7 of 2 M NaNO_3_ Concentration	33

**Table 5 polymers-14-04023-t005:** Consistency value *k* and flow Index value *n* for type1 fluids.

Sr. No.	Fluid	Function Type	*k*	*n*
1	Fluid 1 of 0.2 M NaNO_3_ Concentration	Power function	3.7839	0.359
2	Fluid 2 of 0.4 M NaNO_3_ Concentration	Power function	9.1389	0.297
3	Fluid 3 of 0.6 M NaNO_3_ Concentration	Power function	8.9884	0.363
4	Fluid 4 of 0.8 M NaNO_3_ Concentration	Power function	9.7143	0.36
5	Fluid 5 of 1 M NaNO_3_ Concentration	Power function	7.7744	0.396
6	Fluid 6 of 1.5 M NaNO_3_ Concentration	Power function	6.4291	0.414
7	Fluid 7 of 2 M NaNO_3_ Concentration	Power function	6.8167	0.381

**Table 6 polymers-14-04023-t006:** Consistency value *k* and flow Index value *n* for type2 fluids.

Sr. No	Fluid	Function	*k*	*n*
1	Fluid 1 of 0.2 M NaNO_3_ Concentration	Logarithmic Function	NA	NA
2	Fluid 2 of 0.4 M NaNO_3_ Concentration	Power function	1.534	0.32
3	Fluid 3 of 0.6 M NaNO_3_ Concentration	Power function	11.351	0.394
4	Fluid 4 of 0.8 M NaNO_3_ Concentration	Power function	6.569	0.427
5	Fluid 5 of 1 M NaNO_3_ Concentration	Power function	7.817	0.376
6	Fluid 6 of 1.5 M NaNO_3_ Concentration	Power function	8.257	0.309
7	Fluid 7 of 2 M NaNO_3_ Concentration	Logarithmic Function	NA	NA

**Table 7 polymers-14-04023-t007:** The Factor of optimum fluids of type1 and type2 for pressure drop calculation during laminar flow in a pipe.

Factor	Optimum Fluid of Type1 (without NPs) Fluids	Optimum Fluid of Type2 (with NPs) Fluids
n	0.36	0.394
a	0.58	0.56
b	0.91	1.01
c	0.33	0.40

## Data Availability

Not Applicable.

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
