# Peer review of "A Novel ZnO Nanoparticles Enhanced Surfactant Based Viscoelastic Fluid Systems for Fracturing under High Temperature and High Shear Rate Conditions: Synthesis, Rheometric Analysis, and Fluid Model Derivation"

_polymers, 2022, doi:10.3390/polym14194023_

Round 1
Reviewer 1 Report
In this paper, the authors proposed a viscoelastic fluid system based on cationic cetrimonium bromide surfactants and modified it by adding ZnO nanoparticles. The author argued that adding nanoparticles could improve the rheology of the fluid at the high shear rate and temperature and investigated the performance of different fluids to support their hypothesis. Before this paper could be published, I have several comments,
1. The word "innovative" should be avoided either in the title or in the context of the manuscript.
2. How the range of shear rate and temperature are selected in this paper? Need to be clarified in the introduction or results part.
3. Figures 1 and 2 can be combined into a single figure as they are actually illustrating the same thing.
4. More details, like the name of contents and solutions, and the key processing techniques should be provided in figures 3 and 4.
5. Figure 7 is replotting the data in Figure 6 when C=0.8M? if so, why don't plot Figure 6 in the form of Figure 7? why 0.8M is the optimum concentration?
6. To get a better, comparison, Figures 6, 9, 11, and 12 may be plotted as the ratio between improved and unimproved fluid.
Reviewer 2 Report
I recommend accepting the manuscript after the authors do the major revisions included in the review report in the attachment file.

Round 2
Reviewer 1 Report
The authors have addressed all my concerns. I would like to recommend the acceptance.
Reviewer 2 Report
After reviewing the revised version, it was found that the authors made the required improvements. They also responded to many concerns. We now believe that the manuscript can be accepted for publication in the esteemed Journal. The final decision is yours and the academic editor.